# Integrated Analysis of Methylomic and Transcriptomic Data to Identify Potential Diagnostic Biomarkers for Major Depressive Disorder

**DOI:** 10.3390/genes12020178

**Published:** 2021-01-27

**Authors:** Yinping Xie, Ling Xiao, Lijuan Chen, Yage Zheng, Caixia Zhang, Gaohua Wang

**Affiliations:** 1Department of Psychiatry, Renmin Hospital of Wuhan University, Wuhan 430060, China; yinpingxie@whu.edu.cn (Y.X.); yagezheng2012@sina.com (Y.Z.); 2Institute of Neuropsychiatry, Renmin Hospital, Wuhan University, Wuhan 430060, China; anemous7851@hotmail.com; 3College of Life Sciences, Fujian Agriculture and Forestry University, Fuzhou 350002, China; chen_lijuanrabbit@126.com; 4Department of Cardiology, Renmin Hospital of Wuhan University, Wuhan 430060, China; 13835198286@163.com

**Keywords:** major depressive disorder, diagnostic biomarkers, DNA methylation, mRNA expression, random forest algorithm, leave-one-out cross-validation

## Abstract

Major depressive disorder (MDD) is a mental illness with high incidence and complex etiology, that poses a serious threat to human health and increases the socioeconomic burden. Currently, high-accuracy biomarkers for MDD diagnosis are urgently needed. This paper aims to identify novel blood-based diagnostic biomarkers for MDD. Whole blood DNA methylation data and gene expression data from the Gene Expression Omnibus database are downloaded. Then, differentially expressed/methylated genes (DEGs/DMGs) are identified. In addition, we made a systematic analysis of the DNA methylation on 5′-C-phosphate-G-3′ (CpGs) in all of the gene regions, as well as different gene regions, and then we defined a “dominant” region. Subsequently, integrated analysis is employed to identify the robust MDD-related blood biomarkers. Finally, a gene expression classifier and a methylation classifier are constructed using the random forest algorithm and the leave-one-out cross-validation method. Our results demonstrate that DEGs are mainly involved in the inflammatory response-associated pathways, while DMGs are primarily concentrated in the neurodevelopment- and neuroplasticity-associated pathways. Our integrated analysis identified 46 hypo-methylated and up-regulated (hypo-up) genes and 71 hyper-methylated and down-regulated (hyper-down) genes. One gene expression classifier and two DNA methylation classifiers, based on the CpGs in all of the regions or in the dominant regions are constructed. The gene expression classifier possessed the best predictive ability, followed by the DNA methylation classifiers, based on the CpGs in both the dominant regions and all of the regions. In summary, the integrated analysis of DNA methylation and gene expression has identified 46 hypo-up genes and 71 hyper-down genes, which could be used as diagnostic biomarkers for MDD.

## 1. Introduction

Major depressive disorder (MDD) is one of the most common mental disorders around the world, and as estimated by the World Health Organization, approximately 350 million people of all ages worldwide suffer from depression [1]. The symptoms of depression include pervasive and persistent low mood, lack of motivation, and loss of interest in social interactions, which are important public health problems contributing to severe morbidity and mortality [2,3,4]. At present, the most common classical method for diagnosing depression is scale assessment, and imagological diagnosis may provide effective information in MDD classification [5,6]. Recently, as a relatively low-invasive and accessible method, peripheral blood (PB) examination has become an important complement to the classic diagnostic method mentioned above, improving the accuracy of an MDD diagnosis [7]. For example, microRNAs, exosomes, or a certain protein (e.g., C-reactive protein) in PB samples have been used as powerful tools to distinguish MDD patients from healthy controls [8,9]. To date, high-accuracy biomarkers for MDD diagnosis and prognosis are still lacking. Therefore, it is of great significance to investigate the molecular mechanism of MDD, aiming to identify precise targets and necessary biomarkers for the diagnosis of MDD.

MDD is a complex and heterogeneous disease strongly associated with genetic and environmental factors [10]. Each genetic or environmental factor alone cannot sufficiently explain MDD [11,12]. This then motivates people in the field of MDD research to select epigenetic mechanisms as prime candidates for mediating the genetic and environmental interactions in several brain regions [13]. Furthermore, DNA methylation is one of the major epigenetic modifications and plays an important role in the etiology of complex diseases [14]. Thus far, most DNA methylation studies have used candidate gene approaches and have been predominantly focused on gene promoter regions [15,16,17]. Accompanied by the rise of DNA methylation chip arrays and whole-genome bisulfite sequencing technology, several attempts have been made to decipher the relationship between DNA methylation and depression [18,19]. Although many genome-wide studies have indicated that DNA methylation is associated with depression, both positive and negative associations have been reported, and conflicting results are often observed [20,21]. 

DNA methylation is a very complicated phenomenon. It can occur in different regions such as in transcriptional start sites (TSS), gene bodies, and beyond. Except for the most studied promoter region, the function of other regions has been a largely underexplored domain. Perhaps the average methylation level of a specific gene is not a good reflection of methylation and disease, and a certain region or a CpG site may be better. In this study, we made a systematic analysis of DNA methylation on CpGs in all gene regions, as well as different gene regions (i.e., TSS1500, TSS200, 5′ untranslated region (UTR), first exon (Exon1st), gene body, and 3′UTR), and then defined a “dominant” region, making the DNA methylation research in MDD more elaborate.

Both DNA methylation and gene expression are associated with depression, and conventional wisdom holds that DNA methylation has a negative regulatory effect on gene expression. Their combined analysis gives us a more in-depth understanding of MDD. Hence, we aimed to screen for genes associated with methylation alterations, as well as gene expression changes, through integrated analysis to provide more accurate diagnosis biomarkers for MDD. In this study, integrated analysis of DNA methylation and gene expression data identified 46 hypo-methylated and up-regulated (hypo-up) genes and 71 hyper-methylated and down-regulated (hyper-down) genes, and the random forest (RF) algorithm and leave-one-out (LOO) cross-validation method indicated that they could be used as diagnostic biomarkers for MDD.

## 2. Materials and Methods 

### 2.1. Data Collection

The Gene Expression Omnibus (GEO) database is the largest and most comprehensive public gene expression data resource archiving and sorting high-throughput gene expression and genomics data. Our goal was to identify diagnostic biomarkers through a relatively accessible and low-invasive method. Therefore, we used the keywords “MDD” or “major depressive disorder” and “blood sample” for retrieval from GEO. After a careful review, the DNA methylation dataset GSE113725, including blood samples of 100 MDD patients and 50 healthy controls, was downloaded from the GEO database (GPL13534 Illumina HumanMethylation450 BeadChip) [22]. In addition, a gene expression profile, GSE98793, deposited by Leday et al., was selected based on the GPL570 HG-U133_Plus_2 platform, containing blood samples of 128 MDD patients and 64 healthy controls [23]. The demographic and clinical features for GSE113725 and GSE98793 are listed in Appendix A. The workflow of this study is shown in Figure 1. 

### 2.2. Screening of Differentially Expressed Genes (DEGs)

The “Affy” package in the R software (version 3.5.2) was adopted to process the raw data in CEL format. After eliminating batch differences and performing data background correction, normalization, and summarization, a robust multiarray average was created for further analysis. The “limma” package was applied to assess the differential expression between MDD patients and healthy controls [24]. Benjamini and Hochberg’s (BH) method was used to control the false discovery rate across all genes. The threshold for identifying of DEGs was set at a BH-adjusted *p*-value of <0.05 and a | Log2 fold-change| > 0.2.

### 2.3. Differential Methylation Analysis

Illumina Infinium Human Methylation450 BeadChip array covering 99% of the genes’ annotated promoter (TSS1500, TSS200), 5′UTR, Exon1st, gene body, and 3′UTR in the RefGENE database is one of the most classic DNA methylation detection techniques. TSS200/TSS1500 stands for 0-200/200-1500 bases upstream of the TSS, while gene body refers to the region between the initiation codon and stop codon. The analysis process is as follows:(1) **Methylation level of each CpG.** The methylation level of each CpG can be calculated by the equation β = M / (M + U + a), where *M* > 0, *U* > 0, and *a* ≥ 0. *M* and *U* denote the number of methylated and unmethylated probes, respectively. Since M and *U* are small, “a” is set to 100 to stabilize the β-value [25].(2) **Methylation level of different regions.** In this study, we employed the “ChAMP” package (version 2.18.3) to measure the methylation level of the different regions (TSS1500, TSS200, 5′UTR, Exon1st, gene body, and 3′UTR) for each individual gene using the average β-value of the CpGs in the corresponding regions.(3) **Methylation level of an individual gene.** We also measured the methylation level of an individual gene using the average β-value of the CpGs in all regions.(4) **Identification of differentially methylated genes (DMGs).** To measure the methylation difference between MDD patients and healthy controls, a linear model was built. Ten quantiles of the delta beta value of all genes and all intergenic CpG sites were calculated, and the DMGs were defined as a ∆β value of <1/10 quantile or >9/10 quantile and a BH-adjusted *p*-value of <0.05.

### 2.4. Identification of the **Dominant** Hypo/Hyper-Methylated Regions

In this study, we defined the dominant hypo/hyper-methylated regions (hereafter, dominant regions). A dominant region refers to the smallest delta beta value between the MDD patients and healthy controls. It is worth noting that there may be more than one dominant region in an individual gene. Herein, if the difference between the delta beta value of another region and the smallest delta beta value was smaller than 0.005, the region was regarded as one of the dominant regions [26].

### 2.5. GO and KEGG Enrichment Analyses 

To better understand the biological functions of the DEGs and DMGs, GO enrichment analysis was performed to provide structured annotations on three subontologies: Biological process (BP), molecular function (MF), and cellular component (CC). KEGG pathway enrichment analysis of the DEGs and DMGs was also implemented. Both enrichment analyses employed the R package “clusterprofiler”. A BH-adjusted *p*-value of <0.05 was set as the cut-off criterion.

### 2.6. Classifier Construction and LOO Validation

RF machine learning is a nonlinear classifier that trains a large number of decision trees and uses the class predicted the most from these trees as the final prediction, which has been widely used in bioinformatics analysis, such as in in vivo transcription factor-binding prediction [27], and enhancer identification [28]. For this study, RF, implemented by the R package “randomForest,” was employed to build prediction models to distinguish MDD patients from healthy controls on 46 hypo-up genes and 71 hyper-down genes. Three types of classifiers were trained based on the log2 transformed gene expression level data, the average β-value of the CpGs in all of the regions, and the average β-value of the CpGs in the dominant regions.

### 2.7. LOO Cross-Validation

LOO is a cross-validation method that removes only one sample from the training set, and each learning set is created by taking all of the samples except one (test set left out). We employed the LOO cross-validation method to monitor the performance of the classifiers using the R package “caret.” The discriminative ability of each classifier was measured by receiver op-erating characteristic (ROC) curves, and the area under the ROC curve (AUROC) was calculated using the R package “ROCR.”

## 3. Results

### 3.1. Identification of the DEGs in MDD 

To identify the DEGs in the MDD patients and healthy controls, we selected the most frequently used dataset, namely, GSE98793, containing blood samples of 128 MDD patients and 64 healthy controls. By employing the linear modeling approach, a total of 1506 DEGs were identified, 713 of which were up-regulated and the other 793 down-regulated (Figure 2A and Appendix A). The top 50 genes with significant differences in up- and down-regulated genes were selected to construct a heat map to show the changes in the DEG expression (Figure 2B). 

### 3.2. GO and KEGG Enrichment Analysis of the DEGs

To explore the biological relevance of the DEGs, we performed GO and KEGG enrichment analysis and found that the DEGs were associated with the following: (1) BP terms: Neutrophil-mediated immunity, neutrophil activation involved in immune response, antimicrobial humoral response, etc.; (2) CC terms: Specific granule lumen, secretory granule lumen, cytoplasmic vesicle lumen, etc.; (3) MF terms: Serine-type endopeptidase activity, serine-type peptidase activity, hydrolase activity, acting on acid phosphorus–nitrogen bonds, etc.; (4) KEGG pathways: Hematopoietic cell lineage, asthma, complement, and coagulation cascades, etc. (Figure 3A–D). These results tie in well with previous studies, wherein inflammation was shown to trigger depression [29,30,31]. Proinflammatorycytokines, including IL-1, IL-6, and TNF-α, exhibited higher circulating levels in MDD patients than in non-depressed individuals [32]. The results of our analysis may provide new inflammation-associated diagnostic biomarkers for depression.

### 3.3. Identification and GO/KEGG Enrichment Analysis of the DMGs 

We identified the significant DMGs based on the linear modeling approach and the delta β value of the CpGs of all of the regions. A total of 8313 DMGs were identified, including 4636 hyper-methylated genes and 3677 hypo-methylated genes (Appendix A). We also performed GO and KEGG enrichment analysis of all of the DMGs, and the results showed that the DMGs were associated with following: (1) BP terms: Axonogenesis, neuron projection guidance, axon guidance, etc.; (2) CC terms: Cell leading edge, synaptic membrane, cell–substrate junction, etc.; (3) MF terms: Actin binding, DNA-binding transcription activator activity, DNA-binding transcription activator activity, RNA polymerase II-specific, etc.; (4) KEGG pathways: Axon guidance, MAPK signaling pathway, Rap1 signaling pathway, etc. (Figure 4A–D). The results highlighted that the circulating DNA methylation probably participates in neurodevelopment and neuroplasticity, which are significantly associated with MDD.

### 3.4. Integrated Analysis of the Gene Expression and DNA Methylation

It is generally considered that there is a negative regulatory relationship between DNA methylation and gene expression. Therefore, we obtained the overlap of hypo-methylated and up-regulated genes, as well as hyper-methylated and down-regulated genes. As a result, 46 hypo-up genes and 71 hyper-down genes were identified (Figure 5A,B). All of the gene symbols are listed in Table 1. The heat map showed the changes in the hypo-up and hyper-down genes between the MDD patients and healthy controls (Figure 5C,D). 

The 46 hypo-up genes were involved in such pathways (Appendix A) as PI3K–Akt signaling pathway, the IL-17 signaling pathway, axon guidance, and neuroactive ligand–receptor interaction. The 71 hyper-down genes were involved in pathways (Appendix A), such as the NF-kappa B signaling pathway, the MAPK signaling pathway, neuroactive ligand–receptor interaction, and the synaptic vesicle cycle. It is worth noting that most of the 46 hypo-up genes and 71 hyper-down genes were associated with both the inflammatory response and neuroplasticity. We also found that a proportion of the 71 hyper-down genes were engaged in nucleotide excision repair, DNA replication, ribosome, and ribosome biogenesis in the eukaryotes pathways. The results reveal that depression should be accompanied by changes in the metabolism of biological macromolecules, such as DNA and proteins. 

### 3.5. Identification of the DMGs Based on CpGs in the Different Regions 

In a certain gene, DNA methylation occurs in different regions, including TSS1500, TSS200, 1stExon, gene body, 5′UTR, and 3′UTR, as well as other regions, the function of which remains unclear. Genomic annotation of the methylation based on the CpGs in different regions revealed a biased genomic distribution. As shown in Figure 6A, the maximum number of DMGs is was in the gene body region (>3000), followed by the TSS1500 region (>2000), while the 1stExon region had the minimum (<500). The DNA methylation distribution tendency of the overlap of the DEGs and DMGs was identical in that the gene body region had more than 100 DMGs ranking first, with the TSS1500 region coming second with more than 80, and the 1stExon region being last with at least 30 (Figure 6B). In terms of individual regions, the gene body had the highest percentage of DMGs. Unfortunately, little research on the DNA methylation in this region has been reported. 

To get a closer look at the distribution of each region, we divided all of the overlaps of DMGs and DEGs into four groups: Hyper-up group, hyper-down group, hypo-up group, and hypo-down group. The methylation distribution of the four groups in each region is listed in Table 2. Figure 6C shows that the distribution of the CpGs of the hyper-down and hypo-up genes in the different regions is quite similar. The gene body and TSS1500 region remained in the top two, while the least in the hyper-down group was in 1stExon and in the hypo-up group was TSS200. The hyper-up group had the largest number of DMGs, and the DNA methylation distribution of the hypo-down group in the six regions was similar to that of the hyper-up group. We also found that the hyper-up group had the largest number in the gene body region (>60). It has been reported that the methylation of the gene body may have a positive impact on gene expression [33]. The relationship between gene body methylation and gene expression remains to be further elucidated. Our results suggest that region-specific methylation may play a potential role in the diagnosis of depression. 

### 3.6. Classifier Construction and ROC Curve

We employed the RF algorithm and the LOO cross-validation method to construct classifiers based on the gene expression and methylation data of the 46 hypo-up genes and 71 hyper-down genes to distinguish the MDD patients from the healthy controls. There were two types of methylation classifiers for the 46 hypo-up genes and 71 hyper-down genes based on the CpGs in all of the regions and the CpGs in the dominant regions only. The relevant information about the methylation of the CpGs in all regions, as well as the CpGs in the dominant regions, of the 46 hypo-up genes are shown in Appendix A, while for the 71 hyper-down genes, the information is presented in Appendix A. 

#### 3.6.1. The Importance Score of the 46 Hypo-Up Genes and the 71 Hyper-Down Genes in Each Classifier

The “importance” function of the “randomForest” package was used to calculate the average importance of each gene in the six classifiers, and their importance was ranked in descending order. Figure 7A–F shows the importance scores of the top 20 genes in each classifier, and all of the genes and importance scores are listed in Table 3 and Table 4.

#### 3.6.2. Determine the Number of Genes with the Best Predictive Power in Each Classifier

To obtain the best classification predictive power of each classifier, we added the candidate genes into each classifier one-by-one in order of importance. Figure 8A–C shows that the top 25, top 12, and top 23 are the best predictors of the hypo-up gene expression classifier, the gene methylation classifier based on the CpGs in all of the regions, that based on the CpGs in the dominant regions, respectively. Figure 8D–F shows that the top 31, top 2, and top 18 are the best predictors of the hyper-down gene expression classifier, the gene methylation classifier based on the CpGs in all of the regions, and that based on the CpGs in the dominant regions, respectively.

#### 3.6.3. The Predictive Ability of Each Classifier 

The ROC curve shows that the gene expression classifier exhibited the best predictive ability (AUC = 0.964, *p* = 1.1 × 10^−25^) for the hypo-up genes. The predictive ability (AUC = 0.712, *p* = 3.7 × 10^−5^) of the methylation classifier based on the CpGs in the dominant regions performed slightly better than that of the classifier based on the CpGs of all of the regions (AUC = 0.677, *p* = 5.3 × 10^−4^) (Figure 9A–C). In regard to the hyper-down genes, the gene expression classifier still presented the best predictive ability (AUC = 0.9993, *p* = 1.9× 10^−29^). The predictive ability of the two methylation classifiers was similar for the classifier based on the CpGs of all of the regions (AUC = 0.712, *p* = 3.2× 10^−5^) and on the CpGs of the dominant regions (AUC = 0.716, *p* = 2.2 × 10^−5^) (Figure 9D–F). The results reveal that for both the hypo-up and hyper-down genes, the classifier based on the gene expression data possessed the best predictive ability (AUC > 0.95), while the classifier based on the CpGs in the dominant regions had a relatively higher predictive ability than the classifier based on the CpGs in all of the regions.

## 4. Discussion

MDD is a mental disorder with high morbidity, a complicated etiology, and a severe socioeconomic burden, lacking diagnostic biomarkers. By conducting bioinformatics analysis and mining of the GSE98793 gene expression dataset and the GSE113725genome-wide methylation dataset, we obtained the following results. DEGs are mainly enriched in the inflammatory response-associated pathways, while DMGs are mainly enriched in the neurodevelopmental- and neuroplasticity-associated pathways. Through integrated analysis of the gene expression and DNA methylation data, 46 hypo-up genes and 71 hyper-down genes were identified. These genes are mainly involved in immune activation, synaptic development, and DNA repair. Classifiers based on the gene expression and DNA methylation in all regions, as well as in the different regions, were established by the random forest algorithm and the LOO cross-validation method. The results reveal that for both the hypo-up and hyper-down genes, the classifier based on the gene expression data exhibited the best predictive power, while the methylation classifier based on the CpGs in the dominant regions possessed a relatively higher predictive ability than the methylation classifier based on the CpGs in all regions.

Sigmund Freud wrote that “the complex of melancholia behaves like an open wound” [34]. Clinical and translational studies have shown that inflammatory responses are associated with the onset and maintenance of MDD. Glial cells, including microglia and astrocytes, are the primary immune mediators of the brain and respond accordingly to external stimuli. Proinflammatory (TNF-α and IL-1β) and anti-inflammatory (IL-1, IL-10, and TGFβ1) factors are released under stress. Increasing levels of proinflammatory mediators such as IL-1, IL-2, IL-6, and TNF-α have been observed in patients with depression [35]. Herein, the GO enrichment analysis indicated that DEGs are associated with neutrophil-mediated immunity and the neutrophil activation involved in immune response, and the results are consistent with previous studies. A growing body of research shows that inflammation is closely related to depression [36], but the exact molecular mechanisms remain to be elucidated. Based on the results of our analysis, we hypothesize that inflammatory responses are involved in the progression of depression, and circulating inflammatory factors could be potential diagnostic biomarkers for MDD.

Interestingly, different from DEGs, DMGs are enriched in axonogenesis, neuron projection guidance, axon guidance, GO terms, and the MAPK signaling pathways. The postmortem and meta-analyses of magnetic resonance imaging studies indicate that hippocampal volume decreases in patients with depression [37,38]. There are two hypotheses for this phenomenon, namely, the neuroplasticity hypothesis and the neurogenesis hypothesis. The former suggests that stress induces the atrophy of mature neurons in the hippocampus, while the latter suggests that stress decreases the number of newborn neurons and neural precursor cells in the dentate gyrus of the hippocampus. As mentioned in the last paragraph, astrocytes and microglia participate in the inflammatory response in the brain, and they also secrete nutrients, such as BDNF, to nourish neurons. BDNF positively regulates nerve polymorphisms, and BDNF expression is decreased in the hippocampus of depressed patients [39,40]. Research suggests that the methylation level of the BDNF promoter region in MDD patients is increased, while the mRNA expression is decreased [41]. It has also been reported that IL-6 modulates synaptic plasticity [42]. We presume that there could be extremely complex crosstalk among inflammatory response, neuroplasticity, gene expression, and DNA methylation, but the molecular mechanisms remain a mystery.

TSS1500, TSS200, 5′UTR, and 1stExon are all related to transcriptional initiation and can be subsumed into the promoter region. Our results showed that DMGs have the highest distribution in the promoter region, and the gene body region also accounts for a large proportion. Basically, there is one CpG island (CGI) in every 10 base pairs in the human genome. However, the content of CGI surrounding the TSS of protein-coding genes is as high as 60%. The promoter region is the most studied region of DNA methylation, and it is considered that the CGI methylation of the promoter region is a hallmark of inhibiting gene expression. 

Little research has been done on DNA methylation in the gene body region. Benefitting from the development of sequencing technology, we have a more complete understanding of genome-wide DNA methylation modification. Except for the promoter region, many CGIs are distributed in the gene body and intergenic region. Ehrlich et al. [43] found that brain tissue contains some of the highest levels of DNA methylation in the gene body. In the human brain, 16% of all CGIs are methylated, while 98% of the annotated 5′ promoter regions are unmethylated and, surprisingly, CGI methylation in the gene body region is up to 34% [44]. Although the methylation function of the gene body is unclear, we speculate that methylation in the gene body region may be more dynamic than in the promoter region. Evidence suggests that the degree of gene body methylation in dividing cells is positively correlated with gene expression [45]. Unlike the methylation of the promoter region, which inhibits transcription initiation, gene body methylation does not prevent—and may even promote—transcription elongation [33]. Our findings provide new insight into research related to gene body methylation and depression.

We constructed three types of classifiers: A gene expression classifier, a methylation classifier based on the CpGs in all of the regions, and a methylation classifier based on the CpGs in the dominant regions for both the 46 hypo-up genes and the 71 hyper-down genes. The classifier based on the gene expression data exhibited satisfactory predictive ability, with an AUC > 0.95. The predictive ability of the methylation classifier based on the CpGs in the dominant regions was relatively better than that of the methylation classifier based on the CpGs in all of the regions. The relationship between DNA methylation and depression is a controversial topic. The results presented by genome-wide DNA methylation studies are multitudinous [20,21]. A detailed study of a certain region(s) or CpG(s) should better demonstrate the relationship between the two. A post-hoc investigation indicated that FKBP5 intron methylation has a negative correlation with transcription activation in MDD patients [46]. In a prospective analysis of major depressive disorder in adolescent girls, the authors found that all four significant CpGs in NR3C1 were in the gene body region: Two sites were located within a transcription factor-binding site (TFBS) region, one was in a region of open chromatin, and one site associated with an enhancer element [47]. Benjamard et al. [48] reported that in the promoter region of parvalbumin, methylation was significantly increased at CpG2 and decreased at CpG4 in the MDD group compared to the control group. Some alterations of CpGs are limited to specific gene phenotypes. In the SS genotype of 5-HTTLPR, depression is significantly involved with a decrease in methylation levels at CpG21, CpG25, and CpG26 [49]. Studies based on a certain region(s) or CpG(s) level should be the future research trend of the relationship between DNA methylation and MDD.

Although all of the classifiers demonstrated favorable predictive ability, the limitations cannot be ignored. First, the samples of gene expression and DNA methylation data came from different cohorts. Complicated diseases (such as MDD) involve molecular changes at multiple levels, such as at the genome, epigenome, and transcriptome levels. Researchers hope to systematically and comprehensively study the pathogenesis of diseases from multiple dimensions and perspectives. However, due to limited data sources and research funding, many studies have been conducted using datasets of similar disease models or similar research backgrounds. For example, Reference [26] integrated DNA methylation and transcriptome data and identified 85 hypo-up genes that could be potential diagnostic biomarkers for Parkinson’s disease. The same integrated analysis was performed in Reference [50] to predict gastric cancer. Integrated analyses of other omics are also common in medical research, namely, multi-genome [51], and multi-transcriptome [52] analyses. Not only is this phenomenon observed in clinical studies, but also in botanical studies (e.g., multi-transcriptome in References [53,54], and multi-ChIP-seq in Reference [55]). Although our data did not match completely, we used the integrated analysis method of different omics to provide a new perspective and direction for depression. Second, to what extent changes in peripheral blood genes are associated with genes in the brain is unknown, and integrated analysis of peripheral blood samples and brain tissues is necessary. Finally, further experimental validation will improve the credibility of genes in the classifiers as potential biomarkers for MDD.

## 5. Conclusions

For the first time, three types of classifiers (i.e., a gene expression classifier, a methylation classifier based on the CpGs in all of the regions, and a methylation classifier based on the CpGs in the dominant regions) were constructed and compared with one another on the basis of integrated analysis in MDD. The results showed that for the 46 hypo-up genes and the 71 hyper-down genes, the gene expression classifier presented the best predictive power, while the methylation classifier based on the CpGs in the dominant regions was relatively better than the methylation classifier based on the CpGs in all of the regions. Taken together, we identified a blood signature consisting of 46 hypo-up genes and 71 hyper-down genes, which may play a potential role in the diagnosis of MDD.

## Figures and Tables

**Figure 1 genes-12-00178-f001:**
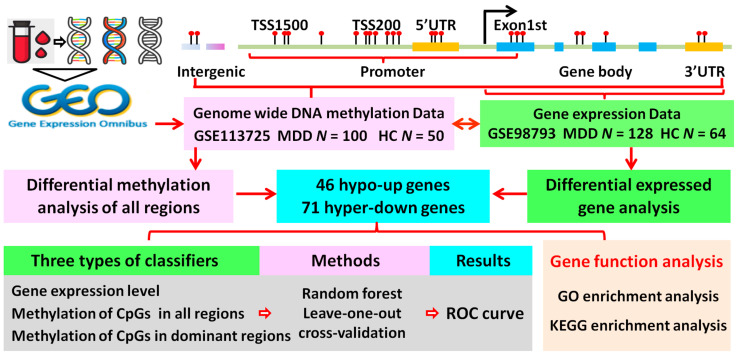
Flowchart of the analysis process. CpGs, 5′-C-phosphate-G-3′; GO, Gene Ontology; HC, healthy control; hyper-down, hyper-methylated and down-regulated; hypo-up, hypo-methylated and up-regulated; KEGG, Kyoto Encyclopedia of Genes and Genomes; MDD, major depressive disorder; ROC, receiver operating characteristic; TSS, transcriptional start site; UTR, untranslated region. *N*, number of MDD or HC.

**Figure 2 genes-12-00178-f002:**
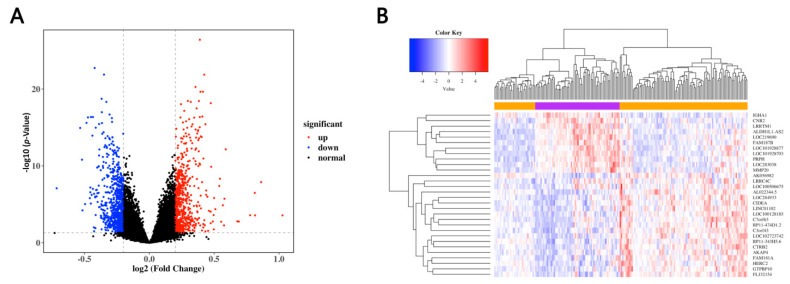
Identification of the DEGs in GES98793. (**A**) Volcano plots of differentially expressed genes (DEGs) in the MDD patients and healthy controls. The red and blue dots represent the up-regulated and down-regulated genes, respectively, while the black dots refer to the non-DEGs. (**B**) Heat map of the top 50 DEGs in the MDD patients and healthy controls.

**Figure 3 genes-12-00178-f003:**
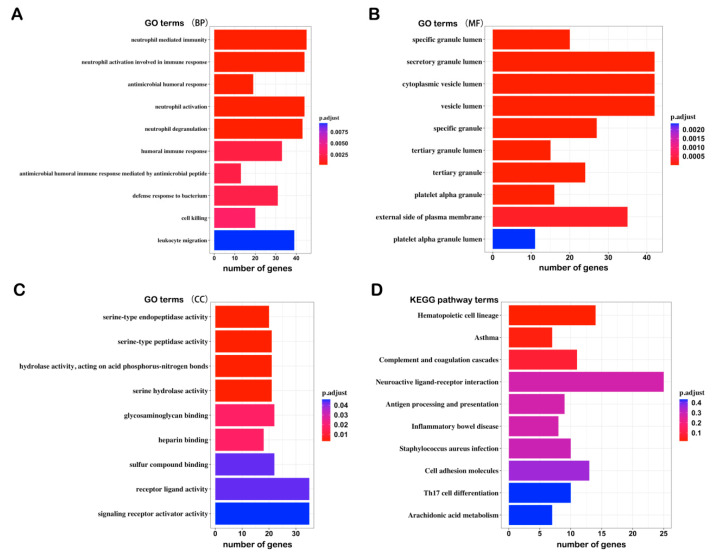
Top 10 GO and KEGG terms from the pathway enrichment analysis of the DEGs: (**A**) Biological process (BP); (**B**) molecular function (MF); (**C**) cellular component (CC); (**D**) KEGG pathway.

**Figure 4 genes-12-00178-f004:**
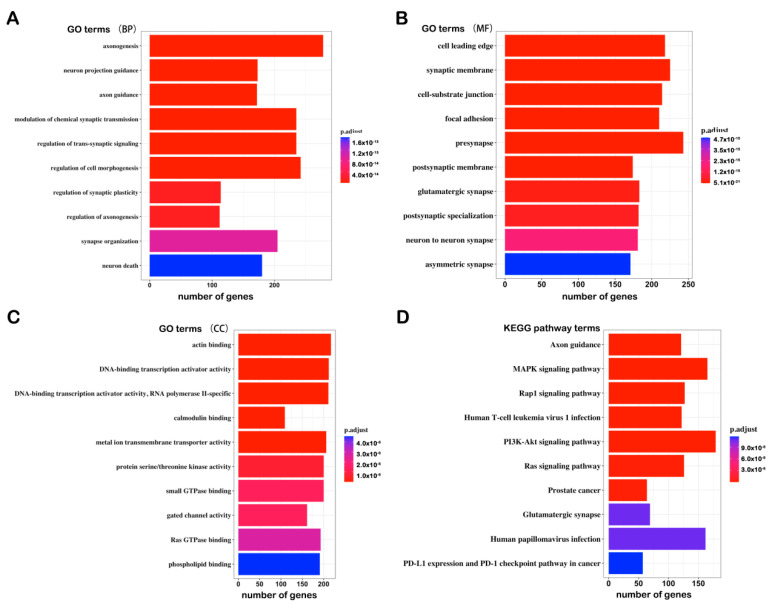
Top 10 GO and KEGG terms of the analysis of the differentially methylated genes (DMGs): (**A**) Biological process; (**B**) molecular function; (**C**) cellular component; (**D**) KEGG pathway.

**Figure 5 genes-12-00178-f005:**
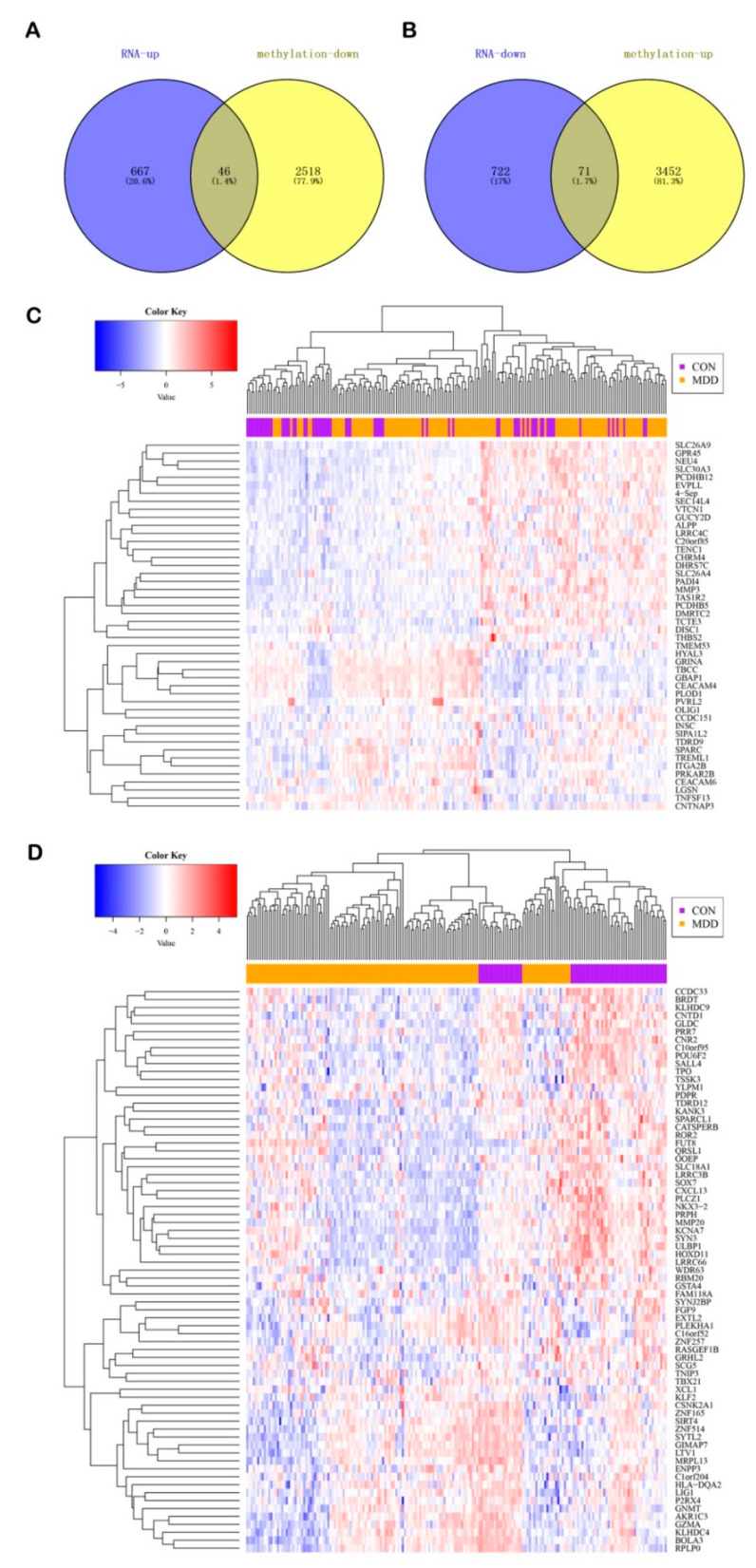
Identification of the hypo-up genes and hyper-down genes. Venn diagram of (**A**) the hypo-up genes and (**B**) the hyper-down genes. Heatmap of (**C**) the hypo-up genes and (**D**) the hyper-down genes between the MDD patients and healthy controls.

**Figure 6 genes-12-00178-f006:**
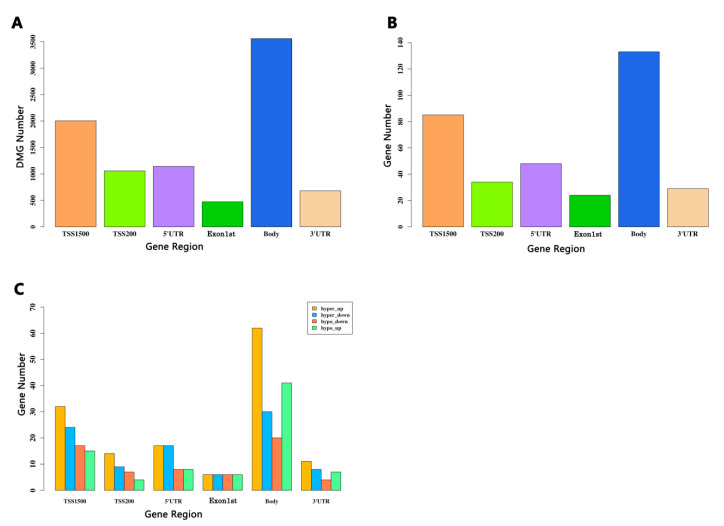
Integrated analysis of the DEGs and DMGs based on the CpGs in different regions. (**A**) Barplot for the CpG distribution of the DMGs. (**B**) Barplot for the overlapped genes between the DEGs and the different regions of the DMGs. The y-axis stands for the overlapped gene numbers. The x-axis represents different gene regions: TSS1500, TSS200, 5′UTR, Exon1st, body, and 3′UTR. (**C**) Barplot of the four groups that overlap in each region. Hyper-up represents the hyper-methylated and up-regulated genes. The y-axis is the number of genes. The x-axis represents different gene regions: TSS1500, TSS200, 5′UTR, Exon1st, body, and 3′UTR.

**Figure 7 genes-12-00178-f007:**
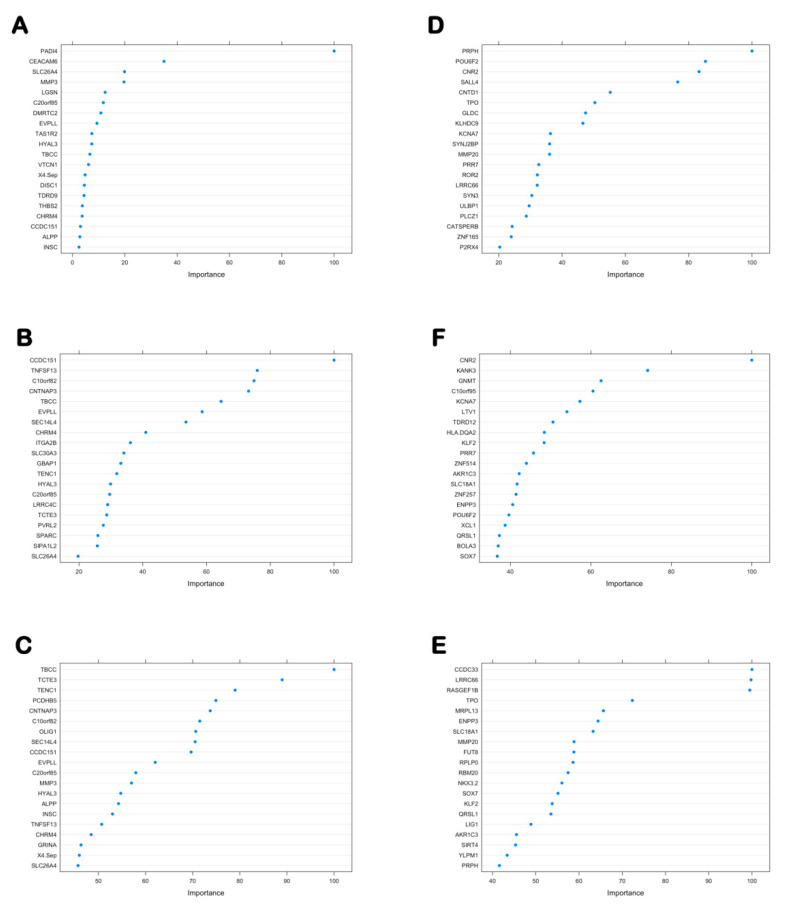
Top 20 genes and importance scores of the hypo-up genes and hyper-down genes in each classifier. Top 20 genes and importance scores of the hypo-up genes in (**A**) the gene expression classifier, (**B**) the gene methylation classifier based on CpGs in all of the regions, and (**C**) the gene methylation classifier based on the CpGs in the dominant regions. Top 20 genes and importance scores of the hyper-down genes in (**D**) the gene expression classifier, (**E**) the gene methylation classifier based on the CpGs in all of the regions, and (**F**) gene methylation classifier based on the CpGs in the dominant regions.

**Figure 8 genes-12-00178-f008:**
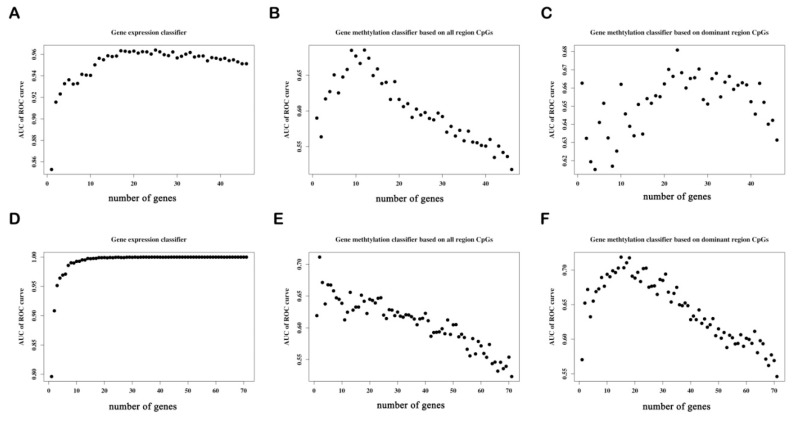
Scatter plots diagrammatizing the relationship between the prediction ability and the number of hypo-up genes and hyper-down genes in each classifier. Classifiers of the hypo-up genes: (**A**) The gene expression classifier, (**B**) the gene methylation classifier based on the CpGs in all of the regions, and (**C**) the gene methylation classifier based on the CpGs in the dominant regions. Classifiers of hyper-down genes: (**D**) The gene expression classifier, (**E**) the gene methylation classifier based on the CpGs in all of the regions, and (**F**) the gene methylation classifier based on the CpGs in the dominant regions. ROC, receiver operating characteristic; AUC, area under the curve; y-axis, AUC value of the ROC curve for the classifier; x-axis, the number of genes in the classifier.

**Figure 9 genes-12-00178-f009:**
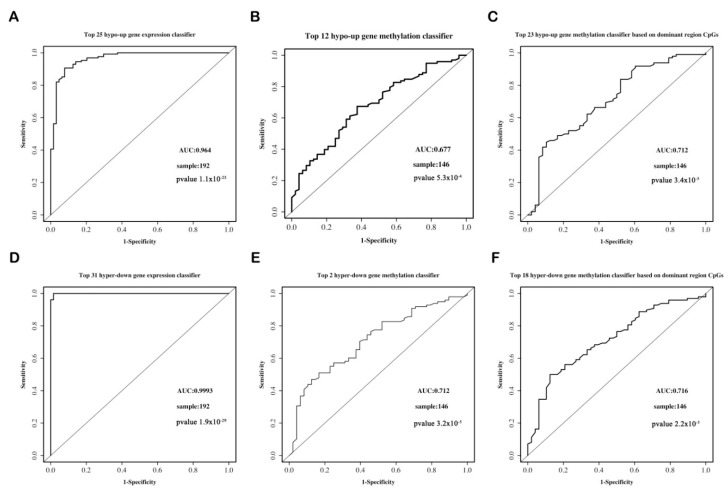
ROC curves for the hypo-up genes and hyper-down gene classifiers. (**A**) Top 25 hypo-up gene expression classifier. (**B**) Top 12 hypo-up gene methylation classifier based on the CpGs in all of the regions. (**C**) Top 23 hypo-up gene methylation classifier based on the CpGs in the dominant regions. (**D**) Top 31 hyper-down gene expression classifier. (**E**) Top 2 hyper-down gene methylation classifier based on the CpGs in all of the regions. (**F**) Top 18 hyper-down gene methylation classifier based on the CpGs in the dominant regions. ROC, receiver operating characteristic; AUC, area under the curve.

**Table 1 genes-12-00178-t001:** Gene symbol of 46 hypo-up genes and 71 hyper-down genes.

Direction	Gene Symbol
hypo-up	*SEPT4, ALPP, C20orf85, CCDC151, CEACAM4, CEACAM6, CHRM4, CNTNAP3, DHRS7C, DISC1, DMRTC2, EVPLL, GBAP1, GPR45, GRINA, GUCY2D, HYAL3, INSCI, TGA2B, LGSN, LRRC4C, MMP3, NEU4, OLIG1, PADI4, PCDHB12, PCDHB5, PLOD1, PRKAR2B, PVRL2, SEC14L4, SIPA1L2, SLC26A4, SLC26A9, SLC30A3, SPARC, TAS1R2, TBCC, TCTE3, TDRD9, TENC1, THBS2, TMEM53, TNFSF13, TREML1, VTCN1*
hyper-down	*AKR1C3, BOLA3, BRDT, C10orf95, C16orf52, C1orf204, CATSPERB, CCDC33, CNR2, CNTD1, CSNK2A1CXCL13, ENPP3, EXTL2, FAM118A, FGF9 FUT8, GIMAP7, GLDC, GNMT, GRHL2, GSTA4, GZMA HLA-DQA2, HOXD11, KANK3, KCNA7, KLF2 KLHDC4, KLHDC9, LIG1, LRRC3B, LRRC66, LTV1, MMP20, MRPL13, NKX3-2, OOEP, P2RX4, PDPR, PLCZ1, PLEKHA1, POU6F2, PRPH, PRR7, QRSL1, RASGEF1B, RBM20, ROR2, RPLP0, SALL4, SCG5, SIRT4, SLC18A1, SOX7, SPARCL1, SYN3, SYNJ2BP, SYTL2, TBX21, TDRD12, TNIP3, TPO, TSSK3, ULBP1, WDR63, XCL1, YLPM1, ZNF165ZNF257, ZNF514*

**Table 2 genes-12-00178-t002:** Gene numbers of four groups in TSS1500, TSS200, 5′UTR, Exon1st, Body, 3′UTR region.

Group	TSS1500	TSS200	5′UTR	Exon1st	Body	3′UTR
hypo-up	15	4	8	6	41	7
hypo-down	17	7	8	6	20	4
hyper-up	32	14	17	6	62	11
hyper-down	24	9	17	6	30	8

**Table 3 genes-12-00178-t003:** Gene symbol and importance scores of 46 hypo-up genes in gene expression classifier and gene methylation classifiers.

Importance of 46 Hypo-Up Genes Based on Gene Expression Level		Importance of 46 Hypo-Up Genes Based on DNA Methylation Level of All Regions		Importance of 46 Hypo-Up Genes Based on DNA Methylation Level of Dominant Regions	
Gene Symbol	Importance	Gene Symbol	Importance	Gene Symbol	Importance
*PADI4*	100.00	*CCDC151*	100.00	*TBCC*	100.00
*CEACAM6*	34.98	*TNFSF13*	75.91	*TCTE3*	88.98
*SLC26A4*	19.87	*C10orf82*	74.91	*TENC1*	79.01
*MMP3*	19.66	*CNTNAP3*	73.18	*PCDHB5*	74.94
*LGSN*	12.47	*TBCC*	64.55	*CNTNAP3*	73.72
*C20orf85*	11.79	*EVPLL*	58.62	*C10orf82*	71.51
*DMRTC2*	10.84	*SEC14L4*	53.55	*OLIG1*	70.65
*EVPLL*	9.34	*CHRM4*	40.94	*SEC14L4*	70.53
*TAS1R2*	7.39	*ITGA2B*	36.11	*CCDC151*	69.67
*HYAL3*	7.34	*SLC30A3*	34.08	*EVPLL*	62.05
*TBCC*	6.63	*GBAP1*	33.11	*C20orf85*	57.94
*VTCN1*	6.12	*TENC1*	31.83	*MMP3*	57.02
*X4.S* *EP*	4.80	*HYAL3*	29.89	*HYAL3*	54.74
*DISC1*	4.52	*C20orf85*	29.60	*ALPP*	54.27
*TDRD9*	4.39	*LRRC4C*	28.99	*INSC*	52.99
*THBS2*	3.75	*TCTE3*	28.70	*TNFSF13*	50.71
*CHRM4*	3.68	*PVRL2*	27.63	*CHRM4*	48.47
*CCDC151*	3.04	*SPARC*	25.92	*GRINA*	46.32
*ALPP*	2.79	*SIPA1L2*	25.75	*X4.S* *EP*	45.95
*INSC*	2.47	*SLC26A4*	19.70	*SLC26A4*	45.69
*GPR45*	2.12	*OLIG1*	19.45	*PVRL2*	44.90
*PCDHB5*	2.09	*GPR45*	19.07	*TDRD9*	39.39
*ITGA2B*	2.01	*PCDHB12*	18.70	*DHRS7C*	38.25
*PLOD1*	2.00	*X4.Sep*	17.39	*PCDHB12*	38.25
*SLC30A3*	1.92	*TAS1R2*	17.30	*PADI4*	35.12
*PRKAR2B*	1.84	*SLC26A9*	17.11	*GBAP1*	34.98
*TMEM53*	1.69	*CEACAM4*	16.23	*CEACAM6*	33.94
*PCDHB12*	1.58	*CEACAM6*	16.17	*LGSN*	32.33
*GUCY2D*	1.52	*VTCN1*	16.08	*GUCY2D*	31.03
*TCTE3*	1.32	*MMP3*	15.67	*PLOD1*	30.28
*CEACAM4*	1.27	*TDRD9*	15.51	*PRKAR2B*	29.78
*TNFSF13*	1.26	*NEU4*	15.48	*NEU4*	29.59
*CNTNAP3*	1.26	*DMRTC2*	14.53	*THBS2*	28.19
*LRRC4C*	1.13	*PADI4*	13.87	*SIPA1L2*	26.04
*SEC14L4*	1.07	*GUCY2D*	12.33	*GPR45*	25.63
*DHRS7C*	1.07	*ALPP*	10.95	*VTCN1*	20.80
*SLC26A9*	1.04	*PLOD1*	10.59	*TMEM53*	18.90
*OLIG1*	0.98	*PCDHB5*	9.36	*DMRTC2*	18.00
*GRINA*	0.96	*THBS2*	9.34	*LRRC4C*	11.03
*PVRL2*	0.83	*PRKAR2B*	8.81	*SLC26A9*	10.08
*TENC1*	0.74	*DHRS7C*	7.63	*ITGA2B*	8.65
*NEU4*	0.59	*GRINA*	7.02	*SPARC*	8.55
*SIPA1L2*	0.53	*TMEM53*	6.53	*CEACAM4*	8.35
*SPARC*	0.43	*DISC1*	3.23	*SLC30A3*	7.98
*TREML1*	0.28	*LGSN*	2.52	*TAS1R2*	5.77
*GBAP1*	0.00	*INSC*	0.00	*DISC1*	0.00

**Table 4 genes-12-00178-t004:** Gene symbol and importance scores of 71 hyper-down genes in gene expression classifier and gene methylation classifiers.

Importance of 71 Hyper-Down Genes Based on Gene Expression Level		Importance of 71 Hyper-Down Genes Based on DNA Methylation Level of All Regions		Importance of 71 Hyper-Down Genes Based on DNA Methylation Level of Dominant Regions	
Gene Symbol	Importance	Gene Symbol	Importance	Gene Symbol	Importance
*PRPH*	100.00	*CNR2*	100.00	*CCDC33*	100.00
*POU6F2*	85.32	*KANK3*	74.12	*LRRC66*	99.79
*CNR2*	83.29	*GNMT*	62.55	*RASGEF1B*	99.50
*SALL4*	76.54	*C10orf95*	60.54	*TPO*	72.35
*CNTD1*	55.23	*KCNA7*	57.28	*MRPL13*	65.64
*TPO*	50.37	*LTV1*	54.06	*ENPP3*	64.41
*GLDC*	47.41	*TDRD12*	50.59	*SLC18A1*	63.26
*KLHDC9*	46.57	*HLA.DQA2*	48.45	*MMP20*	58.84
*KCNA7*	36.31	*KLF2*	48.39	*FUT8*	58.82
*SYNJ2BP*	36.05	*PRR7*	45.75	*RPLP0*	58.62
*MMP20*	36.04	*ZNF514*	43.97	*RBM20*	57.46
*PRR7*	32.64	*AKR1C3*	42.20	*NKX3.2*	56.05
*ROR2*	32.18	*SLC18A1*	41.67	*SOX7*	55.14
*LRRC66*	32.14	*ZNF257*	41.42	*KLF2*	53.79
*SYN3*	30.43	*ENPP3*	40.57	*QRSL1*	53.50
*ULBP1*	29.59	*POU6F2*	39.64	*LIG1*	48.87
*PLCZ1*	28.68	*XCL1*	38.70	*AKR1C3*	45.54
*CATSPERB*	24.22	*QRSL1*	37.29	*SIRT4*	45.32
*ZNF165*	23.93	*BOLA3*	36.98	*YLPM1*	43.38
*P2RX4*	20.28	*SOX7*	36.74	*PRPH*	41.58
*GIMAP7*	19.69	*FGF9*	34.30	*PLEKHA1*	41.24
*NKX3.2*	19.57	*SYTL2*	34.13	*SYNJ2BP*	40.67
*CSNK2A1*	19.37	*TBX21*	33.91	*FGF9*	40.42
*LIG1*	18.73	*MRPL13*	33.58	*KLHDC9*	39.15
*C10orf95*	18.64	*TNIP3*	32.47	*SYTL2*	37.12
*HOXD11*	14.99	*C1orf204*	32.40	*CATSPERB*	35.66
*MRPL13*	14.29	*EXTL2*	32.37	*GSTA4*	31.31
*BRDT*	12.75	*GRHL2*	30.95	*GNMT*	30.86
*SYTL2*	12.32	*MMP20*	30.93	*KANK3*	30.20
*RBM20*	12.02	*RASGEF1B*	30.61	*ROR2*	28.88
*SPARCL1*	11.97	*CATSPERB*	29.19	*P2RX4*	28.65
*ZNF514*	11.36	*C16orf52*	27.04	*CXCL13*	27.97
*GRHL2*	11.27	*SIRT4*	25.34	*HLA.DQA2*	27.92
*SLC18A1*	11.15	*GLDC*	24.13	*BOLA3*	26.26
*GNMT*	10.75	*CXCL13*	24.12	*LTV1*	25.90
*LRRC3B*	10.62	*FUT8*	22.93	*GIMAP7*	25.73
*GZMA*	10.06	*SYNJ2BP*	21.97	*FAM118A*	25.50
*FAM118A*	9.29	*KLHDC9*	21.81	*GRHL2*	24.98
*TDRD12*	9.01	*KLHDC4*	21.53	*CSNK2A1*	24.97
*RPLP0*	8.89	*GIMAP7*	20.03	*HOXD11*	24.96
*CXCL13*	8.53	*TSSK3*	19.54	*BRDT*	24.18
*CCDC33*	8.53	*HOXD11*	18.13	*C10orf95*	23.17
*ZNF257*	8.03	*BRDT*	17.55	*C1orf204*	23.08
*C1orf204*	7.90	*PRPH*	16.54	*XCL1*	23.02
*BOLA3*	7.60	*CSNK2A1*	15.97	*TSSK3*	22.30
*TSSK3*	7.59	*TPO*	15.85	*CNR2*	21.92
*KANK3*	7.51	*NKX3.2*	14.99	*TNIP3*	21.08
*ENPP3*	6.89	*RBM20*	14.11	*WDR63*	20.39
*GSTA4*	6.80	*PDPR*	13.91	*OOEP*	19.69
*AKR1C3*	6.75	*LRRC66*	12.73	*KLHDC4*	19.50
*KLHDC4*	6.28	*SPARCL1*	12.28	*GLDC*	19.26
*EXTL2*	5.85	*CNTD1*	12.28	*SPARCL1*	19.17
*SIRT4*	5.50	*FAM118A*	12.22	*GZMA*	18.74
*FGF9*	5.22	*CCDC33*	11.64	*PLCZ1*	18.74
*SCG5*	4.88	*YLPM1*	11.39	*SCG5*	17.37
*FUT8*	4.51	*RPLP0*	9.51	*POU6F2*	16.12
*OOEP*	4.45	*PLEKHA1*	9.49	*PDPR*	15.66
*TNIP3*	4.44	*LIG1*	9.40	*PRR7*	15.27
*PDPR*	4.43	*LRRC3B*	9.20	*EXTL2*	14.63
*PLEKHA1*	4.29	*ZNF165*	8.61	*ZNF165*	13.83
*SOX7*	3.97	*GSTA4*	7.98	*LRRC3B*	13.70
*C16orf52*	3.90	*SYN3*	7.58	*SYN3*	13.64
*RASGEF1B*	3.56	*SALL4*	7.46	*SALL4*	13.33
*WDR63*	3.54	*SCG5*	6.31	*ULBP1*	12.34
*TBX21*	2.92	*ULBP1*	6.12	*TDRD12*	12.16
*HLA.DQA2*	2.77	*OOEP*	5.27	*ZNF514*	8.22
*KLF2*	2.76	*WDR63*	4.40	*C16orf52*	7.70
*LTV1*	2.35	*PLCZ1*	3.34	*TBX21*	7.52
*QRSL1*	1.46	*GZMA*	2.84	*CNTD1*	7.11
*XCL1*	0.72	*ROR2*	0.52	*KCNA7*	5.55
*YLPM1*	0.00	*P2RX4*	0.00	*ZNF257*	0.00

## Data Availability

Not applicable.

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
