# Peer review of "Integrated Analysis of Methylomic and Transcriptomic Data to Identify Potential Diagnostic Biomarkers for Major Depressive Disorder"

_genes, 2021, doi:10.3390/genes12020178_

Round 1
Reviewer 1 Report
This is a bioinformatic study making use of publicly available studies to identify
diagnostic biomarkers for major depressive disorder associated with both DNA methylation and gene expression changes. The authors identified two datasets, a DNA methylation dataset and a gene expression profile dataset whose analysis showed 46 hypo-up genes and 71 hyper-down genes. Furthermore, the predictive power (area under ROC curve) of gene expression classifier, methylation classifier based on CpGs in all regions and methylation classifier based on of CpGs in dominant regions was 0.964, 0.677 and 0.712, respectively.
I have a series of comments:
- Introduction: the statement “DNA methylation and gene expression are both closely related to MDD and the former has a profound regulatory effect on the latter.” is too nonspecific. The authors should be clearer in presenting the links sustaining their rationale.
- It is not clear how the authors ended up with these two datasets: were there specific inclusion/exclusion criteria?
- The forecast appears properly conducted with RF. I wonder however if the small sample size impacted on the parameters of the prediction.
Reviewer 2 Report
In this work, the authors aimed at identifying markers for MDD using an integrated analysis of DNA methylation and gene expression. To do this, the authors took advantage of publicly available DNA methylation and gene expression datasets from patients diagnosed with depression. Authors identified 46 genes that are upregulated and hypomethylated, and 71 genes that are downregulated and hypermethylated in patients with depression relative to healthy individuals. Although this approach is interesting and novel, there is one major concern that reduces the enthusiasm for this work.
Major concern
Authors utilized the GSE113725 dataset which contains samples from individuals with history of depression, individuals with history of depression and inflammatory disorder, individuals with only an inflammatory disorder, and healthy individuals. Furthermore, this dataset also contains individuals undergoing anti-depressant treatment, and individuals with different ages and sex. Authors aimed at correlating this dataset with GSE98793, which contains individuals with MDD and GAD, individuals with only MDD, and healthy controls. It is unclear if authors controlled for potential confounders such as treatment, age, sex, inflammatory disorder, and GAD. Without this analysis, it is unclear to what extent the identified biomarkers can be used to diagnose MDD. These confounders are not mentioned in the methods or discussion.
Minor concern
The conclusions section needs to be rewritten because 1-3 are a summary of the previous literature. 4 is a summary of the results, this statement needs to change to a conclusion. 5 and 6 can be combined. 7 and 8 are a summary of the results, not a conclusion. 9 is not a conclusion.
Reviewer 3 Report
In "Integrated analysis of methylomic and transcriptomic data to identify potential diagnostic biomarkers for major depressive disorder" by Xie et al, the authors seek to identify genes with alterations in expression and methylation as potential diagnostic biomarkers for major depressive disorder using publicly available data from the Gene Expression Omnibus. They identify genes which are either hypo-methylated/up-regulated or hyper-methylated/down-regulated in MDD cases relative to controls, use random forest modelling to create 3 classifications for these genes, and test the classifications ability to predict cases from controls. The study was well executed, methodologically sound and an interesting read. However, I would offer several suggestions for improvement (mainly minor!):
MAJOR CONCERNS
Methods: I'm not really understanding the dominant hyper/hypo part. I understand it conceptually, but I am not following the explanation of how it is calculated. I think this needs clarification. I am not understanding what the delta symbolises (in the decile deltas in the linear model) and why the smallest delta between MDD and controls is what we are interested in. As to me, delta indicates change, so surely you'd want the biggest change between cases and controls? So if delta is not representative of difference, it needs to be spelt out exactly what the delta deciles are representing so that it is obvious to the reader what the measure of interest is. So heavy clarification between lines 135 and 146.
Results:
There are many more significant DEGs and DMGs than the hypo/up, hyper/down overlap. With genes which are not in the overlapping segment of the pie charts, do they have the same direction of effect (hypo/up, hyper/down) but not statistically significant (e.g. significant upregulation of a gene, but the methylation is hypomethylated, but not significantly different from zero), or is it the opposite direction of effect (hypo/down, hyper/up)? Or a mixture of both? I would be interested in that, as biologically, one expects hypomethylated genes to be upregulated and vice-versa. So if there is a high proportion of genes showing the opposite of that, that is worthy of comment.
MINOR CONCERNS
Abstract:
I find the abstract very information dense and quite technical, rather than a summary. I feel like it needs to be re-written to more clearly state the study aim from the outset. I only really had an idea of what the study was trying to achieve by the time I reached the end of the introduction. So "here is the question we want to ask and why it is important; here is the data we used; here is the method we used; most important results; closing statement." I thought including the AUROC findings in the abstract was quite granular, and it wasn't particularly clear what was being predicted (just that expression did better than methylation).
Introduction:
I think the introduction states conclusions with too much certainty, given the papers cited. For example, the reference for the FMRI work is a sample of 35 people from 2006. Using that reference as the basis for the claim that imageological methods are a classic diagnostic tool is unfounded. If a patient is referred with possible depression, FMRI is not used to confirm this. I think the introduction could be a lot shorter and more focused. So a couple of sentences on the standard why MDD is important and how it is a poorly measured phenotype. Why you think your approach could be beneficial clinically (e.g. could yield therapeutic targets). Are there examples from other phenotypes where your approach has proved fruitful?
It would also be useful to state briefly in the introduction how you intend to answer the questions in your aims.
Methods:
Line 118 "Significantly, DEGs were defined using p value < 0.05 and |Log2 fold-change| > 0.2. " - presumably some kind of multiple testing was applied? It would be worth stating that here. Same for line 151 when stating significance threshold for gene set enrichment analysis.
Out of curiosity, why is it only the first exon that is considered, and not all of the exons (either sequentially or in aggregate)? An explanation of why you are only interested in the first exon would be useful.
What does the "a" denote in the equation: β=M/(M+U+a) ?
Line 153 - use "Random Forest" instead of "RF" on first usage.
Results:
Line 258 - might be useful to explain what is meant by "biased genomic features". Presumably this is referring to an uneven distribution of CpG sites across different genomic features?
Reviewer 4 Report
This study was conducted to screen diagnostic biomarkers associated with both DNA methylation changes and gene expression changes for MDD. The authors have not experimentally determined DNA methylation and gene expression in MDD and heathy subjects, but only conducted bioinformatics analysis and mining of a gene expression data set GSE98793 and genome-wide methylation data set GSE113725, from Gene Expression Omnibus (GEO) database. Integrated analysis of DNA methylation and gene expression identified 46 hypo-up 38 and 71 hyper-down genes in MDD.
My major concern is that DNA methylation data and gene expression data were from different cohorts; therefore, it is big question whether they could be integrated at all. The samples are also not very big for this kind of analysis (100 MDD patients and 50 healthy controls for DNA methylation analysis and 128 MDD patients and 64 heathy subjects for gene expression analysis). The authors should perform a power analysis and calculated the needed sample size.
P-value < 0.05 was considered statistically significant. However, the authors performed multiple comparisons; therefore, a correction for multiple testing should be applied.
There were no demographic and clinical data about the subjects included in the study such as age, gender, race, ethnicity, severity or duration of MDD symptoms, medication received etc. The authors discussed how the antidepressant treatment could be associated with the enzymatic activity of DNMTs and DNA methylation status of certain genes; however, there are no data about the type or dose of the antidepressant drugs the MDD patients received.
The results suggesting the alternations in the inflammatory response and neurodevelopment and neuroplasticity in MDD are not novel, and are already common knowledge.
Minor:
All abbreviations should be written as full term when first time mentioned.
The English language needs revision; there are also some typing errors.
Supplementary tables are just excel tables with data, and are not clear and visually acceptable.
In Figures 3 and 4 it is not clear what is on the x-axis.
Round 2
Reviewer 1 Report
The authors have significantly improved the manuscript along the lines of the reviewers' comments. I have no further suggestions.
Author Response
-
Thank you for your positive comments and valuable time.
Reviewer 2 Report
This paper was significantly improved.
Author Response

(The authors gave the same response as above.)

Reviewer 4 Report
The authors have addressed some but not all reviewers suggestions.
My major concern is still that DNA methylation data and gene expression data were from different cohorts; therefore, it is big question whether they could be integrated at all. In the discussion, the authors should acknowledge this as a major limitation of their study and discuss it in detail.
The authors should perform a power analysis and calculated the needed sample size for their study and not demonstrate the sample size of other studies.
There were still no demographic and clinical data about the subjects included in the study such as age, gender, race, ethnicity, severity or duration of MDD symptoms, medication received, type or dose of the antidepressant drugs the MDD patients received, etc. This important data should be added in the study.
The English language still needs revision.
In Figures 3 and 4 please label the y- and x-axis in the Figures and not in the Figure description.
In my opinion the supplementary Tables should be formatted in Word and not in Excel.
